# Magnetic Resonance Studies of Hybrid Nanocomposites Containing Nanocrystalline TiO_2_ and Graphene-Related Materials

**DOI:** 10.3390/ma15062244

**Published:** 2022-03-18

**Authors:** Nikos Guskos, Grzegorz Zolnierkiewicz, Ewelina Kusiak-Nejman, Aleksander Guskos, Konstantinos Aidinis, Marta Bobrowska, Paweł Berczynski, Agnieszka Wanag, Iwona Pelech, Urszula Narkiewicz, Antoni W. Morawski

**Affiliations:** 1Department of Physics, Faculty of Mechanical Engineering and Mechatronics, West Pomeranian University of Technology in Szczecin, al. Piastow 48, 70-311 Szczecin, Poland; nikos.guskos@zut.edu.pl (N.G.); grzegorz.zolnierkiewicz@zut.edu.pl (G.Z.); aleksander.guskos@zut.edu.pl (A.G.); marta.bobrowska@zut.edu.pl (M.B.); pawel.berczynski@zut.edu.pl (P.B.); 2Department of Inorganic Chemical Technology and Environment Engineering, Faculty of Chemical Technology and Engineering, West Pomeranian of Technology in Szczecin, Pulaskiego 10, 70-322 Szczecin, Poland; ewelina.kusiak@zut.edu.pl (E.K.-N.); agnieszka.wanag@zut.edu.pl (A.W.); iwona.pelech@zut.edu.pl (I.P.); antoni.morawski@zut.edu.pl (A.W.M.); 3Department of Electrical and Computer Engineering, Ajman University, Ajman P.O. Box 346, United Arab Emirates; k.aidinis@ajman.ac.ae

**Keywords:** titania, graphene, magnetic moments, magnetic arrangement, magnetic resonance

## Abstract

Nanocomposites based on nanocrystalline titania modified with graphene-related materials (reduced and oxidized form of graphene) showed the existence of magnetic agglomerates. All parameters of magnetic resonance spectra strongly depended on the materials’ modification processes. The reduction of graphene oxide significantly increased the number of magnetic moments, which caused crucial changes in the reorientation and relaxation processes. At room temperature, a wide resonance line dominated for all nanocomposites studied and in some cases, a narrow resonance line derived from the conduction electrons. Some nanocomposites (samples of titania modified with graphene oxide, prepared with the addition of water or butan-1-ol) showed a single domain magnetic (ferromagnetic) arrangement, and others (samples of titania modified with reduced graphene oxide) exhibited magnetic anisotropy. In addition, the spectra of EPR from free radicals were observed for all samples at the temperature of 4 K. The magnetic resonance imaging methods enable the capturing of even a small number of localized magnetic moments, which significantly affects the physicochemical properties of the materials.

## 1. Introduction

Titania is an interesting material with many applications, e.g., photovoltaics, photocatalysis, filters, electronic instruments, food industry, or construction. The positive features of the material are also related to its low price and chemical resistance. The composites of titania with graphene have recently been intensively studied for their photocatalytic [1,2,3,4,5,6,7,8] and conductive [9,10,11,12,13] properties. Thanks to the conductive properties and magnetic arrangement of titania and its composites, their potential application in spintronic are reported as well [14,15,16,17,18]. Localized magnetic centers and the forming magnetic agglomerates play an important role in the physical properties of the composites.

The EPR/FMR (electron paramagnetic resonance/ferromagnetic resonance) [8,19,20,21,22,23,24,25,26,27,28,29,30,31,32] is suitable for investigating these magnetic centers. Magnetic resonance spectra from different magnetic centers and magnetic correlated spin systems were often observed at higher temperatures. Due to the relaxation processes at lower temperatures, spectra associated with the trivalent titanium ions can be seen as other localized magnetic centers. In graphene oxide (GO) and N-doped reduced graphene oxide (N-rGO), the formation of spatially “isolated” magnetic clusters, free radicals, and a Pauli law component was observed.

Each modification by different methods brings new information about photocatalytic processes to understand them as an application possibility better. As pointed out, magnetic resonance imaging methods capture a small number of localized magnetic moments, significantly affecting the physicochemical properties.

In our previous paper [8], titania nanocomposites modified with reduced graphene oxide (rGO), prepared using the solvothermal method followed by calcination, were discussed. The modification of titania with rGO improved the photocatalytic properties. EPR/FMR measurements showed an occurrence of oxygen defects and ferromagnetic ordering systems.

The aim of the present work was to investigate the magnetic resonance spectra of nanocomposites based on nanocrystalline titania and graphene-related materials. Furthermore, we want to emphasize the exceptional sensitivity of the magnetic resonance method. Using this method, a phase structure can be determined more precisely, which is crucial in photocatalytic, photovoltaic, and spintronic processes [8]. Due to the presence of oxygen defects and trivalent titanium ions, localized magnetic moments in the composites containing titania can significantly influence the physicochemical properties. The magnetic resonance imaging method is very sensitive, enabling the capturing of even a small number of localized magnetic moments.

## 2. Materials and Methods

The nanocomposites named TiO_2_-1ButOH_A, TiO_2_-H_2_O-GO_5%_A, TiO_2_-1ButOH-GO_5%_A, TiO_2_-GO_5%_M, TiO_2_-H_2_O-rGO_5%_A, TiO_2_-1ButOH-rGO_5%_A, and TiO_2_-rGO_5%_M (where *A* means autoclave and *M* means mechanical mixing) were prepared utilizing the hydro- or solvothermal method under elevated pressure conditions (under autogenic pressure). The method of materials preparation and detailed characterization was presented in our previous paper [33]. In brief, the first group of samples was obtained by mechanical mixing of 4 g of starting TiO_2_ with 5 wt.% of GO or rGO compounds (samples denoted with M–mechanical mixing). The second group of samples was hydrothermally/solvothermally treated in a BLH-800 pressure autoclave (Berghof Products + Instruments GmbH, Eningen, Germany) together with 5 cm^3^ of ultra-pure water or butan-1-ol at 200 °C for 4 h and then dried at 105 °C for 24 h to remove residual water and alcohol used as pressurizing agent. Water or butan-1-ol were used as chemicals for the one-step hydrothermal/solvothermal preparation method as the pressurising agents in the preparation method. Heating a fine dry powder of the semiconductor in a pressure autoclave did not generate elevated pressure conditions, i.e., due to very low moisture content in the powdered sample. Therefore, no rinsing step was utilized in the case for the mechanically mixed samples.

Detailed XRD studies of the composites were conducted and discussed in the previous paper [33]. The samples contained almost only the anatase phase; the rutile content was lower than 1%.

The EPR/FMR spectra were recorded using a standard X-band spectrometer type Bruker E500 (9.455 GHz) with a magnetic field modulation of 100 kHz. The magnetic field was scaled with an NMR magnetometer. Samples containing 10–30 mg of the tested substance in the form of a fine powder, without any cement dielectric, were sealed into 4 mm in diameter quartz tubes. All measurements were performed at liquid helium and room temperature.

## 3. Results

Figure 1 and Figure 2 present the magnetic resonance spectra of the nanocomposites TiO_2_-1ButOH_A and TiO_2_-1ButOH-GO_5%_A, respectively, recorded at liquid helium (a) and room (b) temperature.

Comparing Figure 1a with Figure 2a, it can be concluded that the modification of titania with graphene oxide resulted in a tenfold increase of the intensity of the resonance line at the temperature of 4K.

When comparing the spectra recorded at room temperature, some narrowing of the resonance line after the modification with GO can be observed (Figure 1b and Figure 2b). 

All nanocomposites exhibited broad resonance lines at room temperature, not observed at liquid helium temperature. This phenomenon is characteristic of magnetic agglomerations introduced into nonmagnetic matrices [34].

The effective g_eff_ factor can be calculated from the magnetic resonance condition (hν = g_eff_ µ_B_H_r_, where *h* is Planck constant, ν is the microwave frequency, µ_B_ is Bohr magneton, and H_r_ is a magnetic resonance field). The integrated intensity related to the number of magnetic moments involved in the resonance is defined as I_int_ = A·∆Hpp^2^, where A corresponds to amplitude and ∆Hpp is the peak-to-peak line width. The effective g_eff_ factors, integrated intensities, and other resonance spectra parameters for the samples under investigation are shown in Table 1. When two values are shown in the same line, the first value corresponds to the Lorentzian function and the second value to the Gaussian function.

The resonance line for the nanocomposite TiO_2_-1ButOH-GO_5%_A is almost symmetrical and well fitted using the Lorentzian function (Figure 3). The three nanocomposites of titania were modified with reduced GO and treated mechanically or in an autoclave (TiO_2_-H_2_O-rGO_5%_A, TiO_2_-1ButOH-rGO_5%_A, and TiO_2_-rGO_5%_M), the best fitting at room temperature was obtained by using two functions: Lorentz and Gauss (Figure 4).

Additionally, a very narrow symmetric resonance line (Figure 3) arising from free radicals [35,36] was observed for the sample TiO_2_-1ButOH-GO_5%_A. This line measured at RT was also observed for the nanocomposite TiO_2_-H_2_O-GO_5%. The narrow resonance line with the g_eff_ = 2.0026(1) and peak-to-peak line width ΔH = 6.6(1) were the same for both samples. However, the intensity in the sample washed with butan-1-ol was over one order greater. Table 1 shows FMR parameters at room temperature, where spectra originating from magnetic agglomerates are more intensiv. In liquid helium temperature, the following values were reached: g_eff_ = 1.9999(1) and peak-to-peak linewidth ΔH = 19.9(1) for TiO_2_-H_2_O-GO_5%_A; g_eff_ = 1.9998(1) and peak-to-peak linewidth ΔH = 17.5(1) for TiO_2_-1ButOH-GO_5%_A. In liquid helium temperature, the amplitude ratio decreased to 1.76 between nanocomposites TiO_2_-H_2_O-GO_5%_A and TiO_2_-1ButOH-GO_5%_A. This may be due to the skin effect. In general, the skin effect is related to an increase of charge density in the surface and subsurface regions of the material, compared to that of the bulk. In our case, the skin effect is associated with an increase in microwave radiation absorption and an increase in conductivity. Our materials are semiconductors that increase conductivity with increasing temperature. Higher conductivity of the nanocomposite treated with butan-1-ol was observed. 

In both kinds of nanocomposites, the essential line width increased at liquid helium temperature. A single Lorentzian function can well fit this line. The experimental EPR parameters, characteristic for localized defect spins, agree with previous reports, e.g., [28,29,35] on the RT EPR spectra of nanocomposites containing modified TiO_2_.

For the nanocomposites containing rGO, very low intensity of the resonance line at liquid helium temperature was observed. 

In TiO_2_/graphitic carbon nanocomposites obtained in thermal treatment at higher temperatures, a wide resonance line was recorded above 150 K [8].

An EPR resonance line was often observed in the multishell nanographite samples and connected with aggregation processes. Therefore, the spectral experimental phenomena observed in this work behave differently than those described in previous works [8,37,38].

In Table 1, the obtained parameters of the magnetic resonance spectra at RT were given. 

The integrated intensity for TiO_2_-1ButOH_A, TiO_2_-H_2_O-rGO_5%_A, TiO_2_-1ButOH-rGO_5%_A, and TiO_2_-rGO_5%_M nanocomposites showed that the most spin systems were found in these four samples. The spectrum for the first, the TiO_2_-1ButOH_A sample, is well fitted using the Gauss function, while for the other three nanocomposites, the spectra are fitted using two functions, Gauss and Lorentz.

In the remaining three nanocomposites with GO (TiO_2_-H_2_O-GO_5%_A, TiO_2_-1ButOH-GO_5%_A, and TiO_2_-GO_5%_M), the quantity of the spin systems was reduced by one order of magnitude. In this case, the best fitting of the resonance spectrum was obtained using Lorentz’ or Gauss’ function. In the three nanocomposites containing rGO, the presence of magnetic anisotropy was determined, and the total integrated intensity increased significantly compared to the first titania sample shown in Table 1. The total integrated intensity (the sum of values mentioned in the fourth column of Table 1) increased 1.57 times for the sample TiO_2_-H_2_O-rGO_5%_A, 4.09 times for the sample TiO_2_-1ButOH-rGO_5%_A, and 3.03 times for the sample TiO_2_-rGO_5%, respectively (in comparison with the first sample). The increase in the total integrated intensity is related to the increase in spin systems involved in magnetic resonance. The reduction of graphene oxide causes a significant increase in the number of magnetic moments and, at the same time, an increase of magnetic anisotropy. It achieves the highest value of ΔH_r_ for the sample TiO_2_-H_2_O-rGO_5%_A (the lowest number of magnetic moments) and the smallest value for theTiO_2_-1ButOH-rGO_5%_A nanocomposite (the highest number of magnetic moments). Such a significant increase in the total integrated intensity in the TiO_2_-1ButOH-rGO_5%_A nanocomposite can be caused by the smaller size of the magnetic agglomerates in this sample.

For the second, third, and fourth samples in Table 1, the line width was much lower than the other four samples. Among the four remaining samples, the lowest number of magnetic moments occurred for the TiO_2_-1ButOH_A nanocomposite, for which the lowest value of the line width was noted. The broadening of the magnetic resonance line is related to the effects of magnetic dipole−dipole interaction. Increasing the number of correlated spin systems results in the development of more significant relaxation processes for the transition from an excited to a normal state. The g_eff_ parameter is associated with the position of the resonance lines, and its change is bound by the resonance condition by changing the internal magnetic field. In the first sample, TiO_2_-1ButOH_A, we can observe the formation of the highest value of the internal magnetic field and the highest line width compared with the next three samples in Table 1. The sample can be taken as a reference to other nanocomposites. In the last three nanocomposites (listed in Table 1) containing rGO, we can observe a significant increase in the number of magnetic agglomerations. It can enhance the magnetic anisotropy, with a significant increase in the internal magnetic field and the width of the resonance linewidth, due to dipole−dipole interactions (Table 1).

A complicated EPR spectra were observed at a low temperature, arising from different magnetic centers in some of the investigated nanocomposites related to defects associated with trivalent titanium ions. They are assigned to surface and bulk oxygen vacancy-stabilized trivalent titanium ions in the lattice sites in the subsurface layer.

The resonance line for the nanocomposites of TiO_2_ with reduced graphene oxide (rGO), prepared using the solvothermal method followed by calcination (samples TiO_2_-H_2_O-rGO_5%_A and TiO_2_-1ButOH-rGO_5%_A), was fitted with the Gaussian and Lorentz functions. The results obtained for the sample TiO_2_-H_2_O-rGO_5%_A showed some significant differences in the positions of the resonant lines, linewidth, and integrated intensities in relation to the other two nanocomposites containing rGO (TiO_2_-1ButOH-rGO_5%_A and TiO_2_-rGO_5%_M).

The magnetic anisotropy was very similar, compared to other nanocomposites described in the previous work [8]. However, comparing the magnetic resonance parameters of the nanocomposites TiO_2_-H_2_O-rGO_5%_A and TiO_2_-1ButOH-rGO_5%_A with the TiO_2_/rGO-8-600 nanocomposite [8], significant differences in the position of the resonance lines and a significant increase in the line width of the resonance lines could be observed.

The preparation method and the additional factor associated with the titania-to-rGO ratio significantly affect those parameters related to reorientation and relaxation processes. Oxygen vacancies play a crucial role in photocatalytic processes, even if localized magnetic moments are present.

A very intense, narrow symmetric resonance line was observed for the nanocomposites TiO_2_-1ButOH-GO_5%_A and TiO_2_-H_2_O-GO_5% at RT with the g_eff_ = 2.0026(1), which could arise from electron conductivity [37]. The broad resonance line observed at RT comes from magnetic agglomerates formed by oxygen defects associated with titanium ions [8]. Complex magnetic systems are formed in the modified titanium dioxide [38]. The XRD method is less sensitive to studying this effect than the EPR/FMR method. In the case of some of TiO_2_/rGO nanocomposites, the magnetic resonance line from magnetic ordering was observed, and in an appropriate calcination temperature system, the integrated intensity of the resonance lines from oxygen defects (free radicals) drastically increased [6].

The nanocomposites titania/rGO exhibited an enhanced photoactivity, e.g., of acid blue or phenol photodecomposition process [8,37].

After separating nanocrystallites, the above composites can be applied in medicine. In addition, considering their nanosize, they can be useful in spintronics, similar to titania doped with metals [16,18,38]. Finally, the nanocomposites can also be used as nano-level magnets because nanopores are usually larger than magnetic nanoparticles.

## 4. Conclusions

All studied nanocomposites showed the presence of magnetic agglomerates. The parameters of magnetic resonance strongly depended on the modification processes. Oxygen reduction significantly increased the number of magnetic moments, which caused crucial changes in the reorientation and relaxation processes. Many localized magnetic centers were observed at low temperatures because resonant lines from magnetic agglomerates are usually more intense at high temperatures. In titania modified with various forms of graphene, the localized ferromagnetic ordering was observed at room temperature. The nanocomposites can also be used in spintronics, thanks to the coexistence of ferromagnetic ordering and electric conduction. The formation of magnetic agglomerates can adversely affect the ordering processes of magnetic moments. Localized oxygen vacancies magnetic centers can essentially influence catalytic and photocatalytic processes.

## Figures and Tables

**Figure 1 materials-15-02244-f001:**
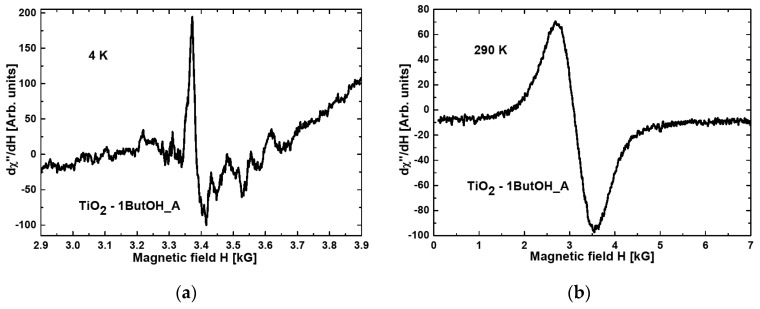
Magnetic resonance spectra obtained at (**a**) 4 K and (**b**) RT for the nanocomposite TiO_2_-1ButOH_A.

**Figure 2 materials-15-02244-f002:**
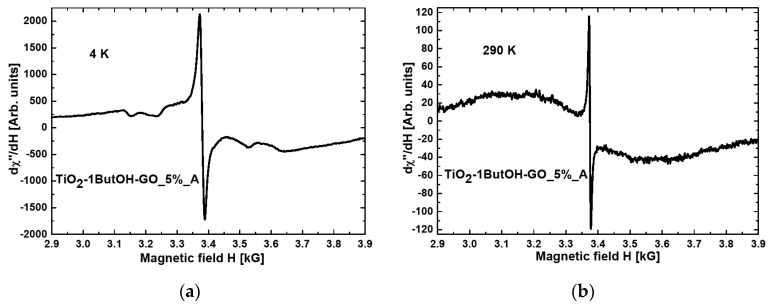
Magnetic resonance spectra measured at (**a**) 4 K and (**b**) RT for nanocomposite TiO_2_-1ButOH-GO_5%_A.

**Figure 3 materials-15-02244-f003:**
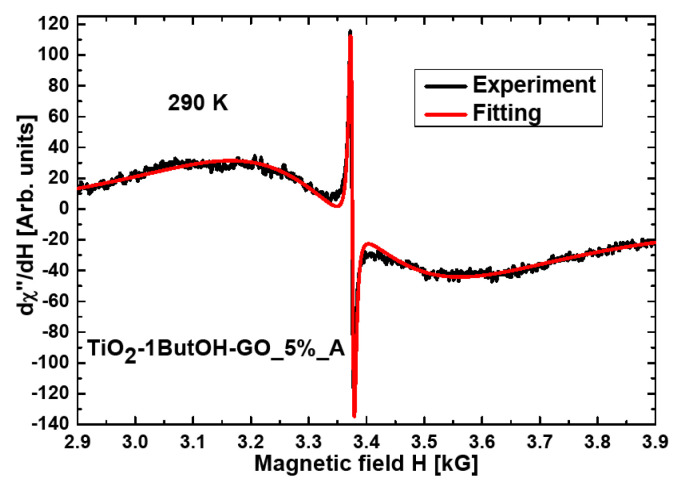
The fitting by the Lorentzian function obtained for the nanocomposite TiO_2_-1ButOH-GO_5%_A.

**Figure 4 materials-15-02244-f004:**
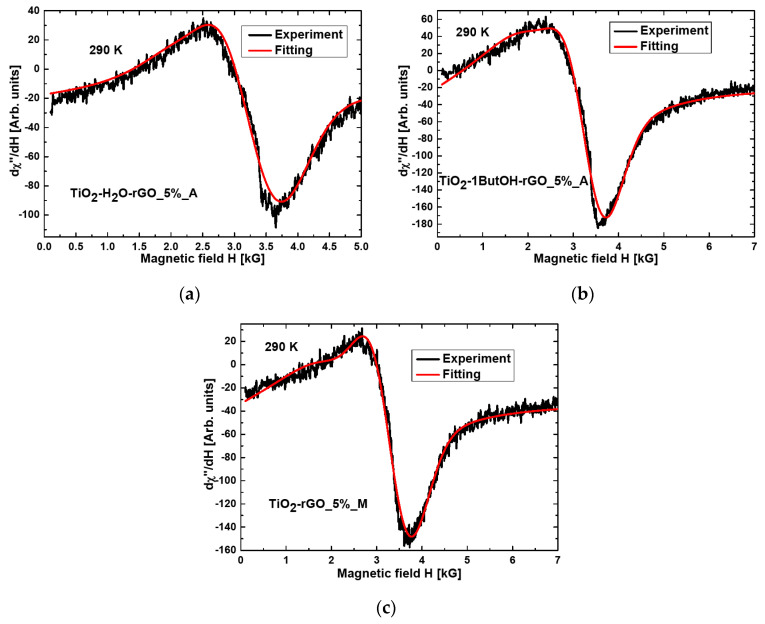
The fitting by Lorentzian and Gaussian function obtained for the nanocomposite (**a**) TiO_2_-H_2_O-rGO_5%_A; (**b**) TiO_2_-1ButOH-rGO_5%_A; (**c**) TiO_2_-rGO_5%_M.

**Table 1 materials-15-02244-t001:** The parameters: g_eff_, line width, integrated intensity, and the difference between resonance fields for modified nanocomposites TiO_2_ at room temperature.

Nanocomposites	g_eff_	ΔH_pp_ (G)	I/I_1_	ΔH_r_ (G)
TiO_2_-1ButOH_A	2.162(1)	912(3)	1.00	
TiO_2_-H_2_O-GO_5%_A	2.025(1)	537(3)	0.07	
TiO_2_-1ButOH-GO_5%_A	2.010(1)	397(3)	0.09	
TiO_2_-GO_5%_M	2.013(1)	336(1)	0.11	
TiO_2_-H_2_O-rGO_5%_A	3.038(90) 2.109(9)	1138(63) 1116(22)	0.36 1.21	980
TiO_2_-1ButOH-rGO_5%_A	2.644(14) 2.065(3)	1701(22) 970(10)	2.85 1.24	717
TiO_2_-rGO_5%_M	2.669(17) 2.046(2)	1838(28) 978(8)	1.94 1.09	772

## Data Availability

The data supporting reported results can be delivered on demand by the authors.

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
