# Peer review of "Magnetic Resonance Studies of Hybrid Nanocomposites Containing Nanocrystalline TiO2 and Graphene-Related Materials"

_materials, 2022, doi:10.3390/ma15062244_

Round 1
Reviewer 1 Report
Here the authors report on the magnetic properties of titanium dioxide nanomaterials modified with graphene-related materials. The authors explore the use of graphene oxide (GO) and reduced graphene oxide (rGO) mixed at 5%, as well as washing with water or butanol, and preparing via autoclave or mechanical mixing. While there were clear trends observed between samples treated with GO vs rGO, there was little discussion surrounding any of the other processing steps (washing solvent or autoclave vs mechanical mixing). While this manuscript should be published, addressing that above along with addition questions/comments below is prior to publication is necessary.
Page 2, Line 46: Statement is incomplete
Experimental section is significantly lacking detail for the composite preparation. The reference to article #33 is useful in part but also requires the reader to seek out yet another reference to get all steps in the preparation. Key steps should still be included in this manuscript.
- In addition, it would be useful to know how many samples of each type were prepared and analyzed. This is useful in supporting whether the trends observed are just a fluke or actually reproducible
What was the purpose or reason for exploring the water washing vs the butanol washing? Why were those solvents chosen over other solvents? Would other solvents be worth investigating? It may be useful to discuss why certain variables were investigated in the introduction or in the discussion of the results. For samples prepared by mechanical mixing, were they washed? If so, was it with water or butanol?
Page 4: The discussions surrounding geff, ΔHpp, and I values starting on page four do not always line up with what is actually listed in Table 1. For example, on page 4, line 104, it mentioned a geff = 2.0026 for the second and third sample do not match the values in Table 1 (2.025 and 2.010, respectively). It also states the intensity of the ButOH-washed sample was over one order greater but it is unclear what Table value this correlates with as I/I1 does not show this.
Page 4, line 110: What is meant by the 'skin effect'?
Author Response
Dear Reviewer,
Thank you very much for your effort and time devoted to our manuscript.
We do agree with all your comments, and we revised the manuscript accordingly.
Here the authors report on the magnetic properties of titanium dioxide nanomaterials modified with graphene-related materials. The authors explore the use of graphene oxide (GO) and reduced graphene oxide (rGO) mixed at 5%, as well as washing with water or butanol, and preparing via autoclave or mechanical mixing. While there were clear trends observed between samples treated with GO vs rGO, there was little discussion surrounding any of the other processing steps (washing solvent or autoclave vs mechanical mixing). While this manuscript should be published, addressing that above along with addition questions/comments below is prior to publication is necessary.
Page 2, Line 46: Statement is incomplete.
Response: It was corrected.
Experimental section is significantly lacking detail for the composite preparation. The reference to article #33 is useful in part but also requires the reader to seek out yet another reference to get all steps in the preparation. Key steps should still be included in this manuscript.
- In addition, it would be useful to know how many samples of each type were prepared and analyzed. This is useful in supporting whether the trends observed are just a fluke or actually reproducible
What was the purpose or reason for exploring the water washing vs the butanol washing? Why were those solvents chosen over other solvents? Would other solvents be worth investigating? It may be useful to discuss why certain variables were investigated in the introduction or in the discussion of the results. For samples prepared by mechanical mixing, were they washed? If so, was it with water or butanol?
Response: More information about the sample preparation was added in the revised version. The wrong information about washing the samples with water of 1-butyl alcohol was given in the first version of the manuscript (corrected in the revised version). There was no step of washing included in the procedure of samples preparation. Water or butan-1-ol were used as chemicals for one-step hydrothermal/solvothermal preparation method as the pressurising agents in the preparation method. Heating a dry fine powder of semiconductor in a pressure autoclave did not generate elevated pressure conditions, i.e. due to a very low content of moisture in the powdered sample. Secondly, we found out in our previous work [37. Kusiak-Nejman, E.; Wanag, A; Kowalczyk, Ł.; Kapica-Kozar, J.; Colbeau-Justin, C.; María G. Mendez Medrano, M.G.; Morawski, A.W. Graphene oxide-TiO2 and reduced graphene oxide-TiO2 nanocomposites: Insight in charge-carrier lifetime measurements, Catalysis Today 287 (2017) 189–195.] that butan-1-ol plays a role of holes scavenger suppressing recombination of photoexcited charge carriers in TiO2, enhacing the photocatalytic efficiency. We would like to show the difference between alcohol-treated and non-treated samples under elevated pressure. Due to this fact we used water as pressurising agent for hydrothermal modification. Additionally, we wanted to know the photocatalytic behaviour of samples unmodified in a pressure autoclave. For this purpose we prepared a series of mechanically mixed TiO2/GO or TiO2/rGO samples. No rinsing step was utilized in case for mechanically mixed samples.
Page 4: The discussions surrounding geff, ΔHpp, and I values starting on page four do not always line up with what is actually listed in Table 1. For example, on page 4, line 104, it mentioned a geff = 2.0026 for the second and third sample do not match the values in Table 1 (2.025 and 2.010, respectively). It also states the intensity of the ButOH-washed sample was over one order greater but it is unclear what Table value this correlates with as I/I1 does not show this.
Response: Thank you very much, that’s our mistake, we did not underline clearly enough, that there were two resonance lines – the wide one and the narrow one. We revised the text accordingly.
"For both samples this narrow resonance line with the geff= 2.0026(1) and peak-to-peak linewidth ΔΗ=6.6(1) were the same, however the intensity in the sample washed with butan-1-ol was over one order greater."
"The data presented in Table 1 describe a wide resonance line."
Page 4, line 110: What is meant by the 'skin effect'?
Response: The result of the skin effect is an increase of charge density in the surface and subsurface region of the material, comparing with that of the bulk. In our case an increase in the absorption of microwave radiation and increase of conductivity was observed. The appropriate changes were introduced in the manuscript.
Reviewer 2 Report
The paper is a result of sound scientific work, however in my opinion, the athors need to expand on the underlying mechanisms which result in the observed changes in the EPR spectra. So far the overall tone of the paper is "the sample sythesized in X way rasults in Y changes in the EPR spectra". However, the key question is WHY. I find that it is not desribed adeqately in the text. The authors should put more emphasis on explaining this.
Also one minor comment - the last sentence of the 2nd paragrapgh in the introduction section (the one that ends in line 42) requiers an appropriate citation.
Author Response
Dear Reviewer,
Thank you very much for your work and effort to revise our manuscript.
The paper is a result of sound scientific work, however in my opinion, the athors need to expand on the underlying mechanisms which result in the observed changes in the EPR spectra. So far the overall tone of the paper is "the sample sythesized in X way rasults in Y changes in the EPR spectra". However, the key question is WHY. I find that it is not desribed adeqately in the text. The authors should put more emphasis on explaining this.
Response: Thank you very much for the valuable remark. We added in the manuscript a following explanation:
" A very intense narrow symmetric resonance line was observed for the nacomposites TiO2-1ButOH-GO_5%_A and TiO2-H2O-GO_5% at RT with the geff= 2.0026(1), which could arise from electron conductivity [37]. The wide resonance line observed at RT is coming from magnetic agglomerates formed by oxygen defects associated with titanium ions [8]. In modified titanium dioxide, complex magnetic systems are formed [38]. The XRD method is much less sensitive to study this effect than EPR/FMR method.
Also one minor comment - the last sentence of the 2nd paragrapgh in the introduction section (the one that ends in line 42) requiers an appropriate citation.
Response: Thank you, the citation was added.
"which is crucial in photocatalytic processes [8]."
Reviewer 3 Report
This work presents a study into the magnetic resonance of TiO2 - graphene composites. The motivation for the study is not sufficiently clear, why is this material system chosen?
The phase of titanium dioxide plays a key role in its magentic properties. Authors should show
- Microstructures of composites produced here
- XRD patterns confirming the anatase or rutile phase in TiO2
- Discussion of anatase/rutile phase composition on magnetic properties
The significance of phase on the magnetic performance in titanium dioxide has been discussed in previous work, and should be examined here.
Some conjecture on the utility of the systems made here towards the applications of spintronics. Other studies into TiO2 based spintronics should be cited. e.g.
https://doi.org/10.1016/j.heliyon.2020.e03338
https://doi.org/10.1063/1.4811539
https://doi.org/10.1088/1367-2630/abae87
The synthesis route studied and used should further be explained with regards to the rationale for this method. i.e. why was this method used?
Author Response
Dear Reviewer,
Thank you very much for your effort and time devoted to our manuscript.
This work presents a study into the magnetic resonance of TiO2 - graphene composites. The motivation for the study is not sufficiently clear, why is this material system chosen?
The phase of titanium dioxide plays a key role in its magentic properties. Authors should show
- Microstructures of composites produced here
- XRD patterns confirming the anatase or rutile phase in TiO2
Response: Of course, thank you for the remarks. Nevertheless, the aspects suggested in points 1 and 2 were discussed in detail in our previous works [33,37]. Unfortunatelly, we are not allowed to duplicate the results due to possible copyright infringement of earlier publications. A detailed XRD studies of the composites were conducted and discussed in the paper [33}. The samples contained almost only anatase phase, the content of rutile was lower than 1%.
The significance of phase on the magnetic performance in titanium dioxide has been discussed in previous work, and should be examined here.
Response: The magnetic centers originating from oxygen defects, can be located in both phases, anatase and rutile. The presence of trivalent titanium ions can affect the EPR spectra. The samples described in this paper contained mainly anatase (99%), the content of rutile was very low.
Some conjecture on the utility of the systems made here towards the applications of spintronics. Other studies into TiO2 based spintronics should be cited. e.g.
https://doi.org/10.1016/j.heliyon.2020.e03338
https://doi.org/10.1063/1.4811539 Copper doped titania
https://doi.org/10.1088/1367-2630/abae87
Response: Thank you very much for the remark, the citations were added as 38, 39, 40.
The synthesis route studied and used should further be explained with regards to the rationale for this method. i.e. why was this method used?
Response: Thank you very much for the remark, the synthesis route was described more in depth in the revised version.
Reviewer 4 Report
The manuscript “Magnetic resonance studies of hybrid nanocomposites containing nanocrystalline TiO2 and graphene-related materials” reports an accurate investigation of magnetic properties of different hybrid nanomaterials. The paper is well structured, well written; it is clear, precise, and easy to understand.
Each statement is validated by appropriate bibliographic references.
As reported by the references, the authors possess a good and consolidated knowledge of the subject treated here.
In my opinion the manuscript is suitable for publication in Materials journal after a very minor revision in order to increase the reader’s interest. I do not require any experimental or scientific revision; however, the following suggestions are proposed:
Abstract
The abstract is clear and concise, however the main findings of your research and why they are important should be reported.
Introduction:
In the introduction session, the novelty and significance of the work should be emphasised. In addition, the potential impact of the research and why it is important, compared to other research in this field or previous studies, should be discussed.
Author Response
Dear Reviewer,
Thank you very much for revising our manuscript.
The manuscript “Magnetic resonance studies of hybrid nanocomposites containing nanocrystalline TiO2 and graphene-related materials” reports an accurate investigation of magnetic properties of different hybrid nanomaterials. The paper is well structured, well written; it is clear, precise, and easy to understand.
Each statement is validated by appropriate bibliographic references.
As reported by the references, the authors possess a good and consolidated knowledge of the subject treated here.
In my opinion the manuscript is suitable for publication in Materials journal after a very minor revision in order to increase the reader’s interest. I do not require any experimental or scientific revision; however, the following suggestions are proposed:
Abstract
The abstract is clear and concise, however the main findings of your research and why they are important should be reported.
Response: A final sentence was added in the revised version of the manuscript as follows:
“The magnetic resonance imaging methods enable to capture even a small number of localized magnetic moments, which significantly affect the physico-chemical properties of the materials.”
Introduction:
In the introduction session, the novelty and significance of the work should be emphasised. In addition, the potential impact of the research and why it is important, compared to other research in this field or previous studies, should be discussed.
Response: The following sentences were added to the Introduction part:
“Localized magnetic moments in the composites containing titania, due to the presence of oxygen defects and trivalent titanium ions, can significantly influence the physico-chemical properties. The magnetic resonance imaging methods are very sensitive, enabling to capture even a small number of localized magnetic moments.
Reviewer 5 Report
This work aims to investigate the magnetic resonance spectra of nanocomposites based on nanocrystalline titania and graphene-related materials.
I found the work interesting because it emphasizes the exceptional sensitivity of the magnetic resonance method, which can determine more accurately a phase structure in many domains.
However, I have only one remark:
line 46: the sentence seems to be not complete. The authors should complete thissentence or reformulate it to be more comprehensible.
Author Response
Dear Reviewer,
Thank you very much for your work and time devoted to our manuscript.
The manuscript “Magnetic resonance studies of hybrid nanocomposites containing nanocrystalline TiO2 and graphene-related materials” reports an accurate investigation of magnetic properties of different hybrid nanomaterials. The paper is well structured, well written; it is clear, precise, and easy to understand.
Each statement is validated by appropriate bibliographic references.
As reported by the references, the authors possess a good and consolidated knowledge of the subject treated here.
In my opinion the manuscript is suitable for publication in Materials journal after a very minor revision in order to increase the reader’s interest. I do not require any experimental or scientific revision; however, the following suggestions are proposed:
Abstract
The abstract is clear and concise, however the main findings of your research and why they are important should be reported.
Response: A final sentence was added in the revised version of the manuscript as follows:
“The magnetic resonance imaging methods enable to capture even a small number of localized magnetic moments, which significantly affect the physico-chemical properties of the materials.”
Introduction:
In the introduction session, the novelty and significance of the work should be emphasised. In addition, the potential impact of the research and why it is important, compared to other research in this field or previous studies, should be discussed.
Response: The following sentences were added to the Introduction part:
“Localized magnetic moments in the composites containing titania, due to the presence of oxygen defects and trivalent titanium ions, can significantly influence the physico-chemical properties. The magnetic resonance imaging methods are very sensitive, enabling to capture even a small number of localized magnetic moments.
Round 2
Reviewer 1 Report
The authors adequately addressed the reviewer comments. The only additional suggests are:
1) add in the volume of solvent (water or alcohol) used for the hydro/solvothermal synthesis.
2) final proof reading is also necessary to fix minor grammatical errors throughout (e.g., page 2, line 84 should either read as 'Detailed XRD studies...' or 'A detailed XRD study...').
Author Response
Dear Reviewer,
Thank you very much for your remarks.
I added the volume of the solvent (5 cm3) and corrected gramma errors.
Faithfully yours,
Urszula Narkiewicz